# Living Together, Singing Together: Revealing Similar Patterns of Vocal Activity in Two Tropical Songbirds Applying BirdNET

**DOI:** 10.3390/s24175780

**Published:** 2024-09-05

**Authors:** David Amorós-Ausina, Karl-L. Schuchmann, Marinez I. Marques, Cristian Pérez-Granados

**Affiliations:** 1Urbanización Maryvilla, 03710 Calpe, Spain; daa40@gcloud.ua.es; 2Computational Bioacoustics Research Unit (CO.BRA), Institute for Science and Technology in Wetlands (INAU), Federal University of Mato Grosso (UFMT), Cuiabá 78060-900, Brazil; klschuchmann@googlemail.com (K.-L.S.); marinez513@gmail.com (M.I.M.); 3Ornithology, Zoological Research Museum A. Koenig (ZFMK), 53113 Bonn, Germany; 4Postgraduate Program in Zoology, Institute of Biosciences, Federal University of Mato Grosso, Cuiabá 78060-900, Brazil; 5Conservation Biology Group, Landscape Dynamics and Biodiversity Programme, Forest Science and Technology Center of Catalonia (CTFC), 25280 Lleida, Spain; 6Ecology Department, Alicante University, 03080 Alicante, Spain

**Keywords:** BirdNET, *Campylorhynchus turdinus*, *Cantorchilus leucotis*, confidence score, machine learning, passive acoustic monitoring, troglodytidae, vocal behavior

## Abstract

In recent years, several automated and noninvasive methods for wildlife monitoring, such as passive acoustic monitoring (PAM), have emerged. PAM consists of the use of acoustic sensors followed by sound interpretation to obtain ecological information about certain species. One challenge associated with PAM is the generation of a significant amount of data, which often requires the use of machine learning tools for automated recognition. Here, we couple PAM with BirdNET, a free-to-use sound algorithm to assess, for the first time, the precision of BirdNET in predicting three tropical songbirds and to describe their patterns of vocal activity over a year in the Brazilian Pantanal. The precision of the BirdNET method was high for all three species (ranging from 72 to 84%). We were able to describe the vocal activity patterns of two of the species, the Buff-breasted Wren (*Cantorchilus leucotis*) and Thrush-like Wren (*Campylorhynchus turdinus*). Both species presented very similar vocal activity patterns during the day, with a maximum around sunrise, and throughout the year, with peak vocal activity occurring between April and June, when food availability for insectivorous species may be high. Further research should improve our knowledge regarding the ability of coupling PAM with BirdNET for monitoring a wider range of tropical species.

## 1. Introduction

Owing to the current decline in biodiversity, there is a growing need for automated and effective methods to improve wildlife monitoring [1]. Establishing an effective ecological monitoring methodology is essential for determining changes in species richness and population trends over time, which is needed for proper management and conservation of natural ecosystems and biodiversity. Traditional monitoring methods, such as line transects and point counts, require significant human effort, are limited in space and time, and may be subject to biases and limitations, such as the experience and hearing ability of the observer [2]. For this reason, numerous automated and noninvasive tools have emerged in recent years [3,4,5]. These tools do not require human presence and can help monitor ecological processes more effectively. However, it is important to test the effectiveness of these tools and standardize them before they are implemented on a large scale [1].

Acoustic communication is used by many groups of animals to share information with members of their own or other species. Therefore, monitoring species via acoustic cues is a common method for assessing changes in species abundance, population richness, or community composition, among other factors. However, acoustic surveys are subject to various biases on the basis of experience and the detection and identification capacity of the observer [6]. To avoid such biases, in recent years, a novel noninvasive and automated technique for wildlife monitoring, passive acoustic monitoring (PAM), has emerged [7]. PAM is based on the deployment of Autonomous Recording Units (ARUs) equipped with acoustic sensors (microphones hereinafter), which are deployed in the field and programmed to record at time periods of interest. The posterior analysis of the collected recordings enables monitoring wildlife in an automated manner. In recent years, the use of PAM has rapidly increased in both aquatic and terrestrial environments [7,8]. Among the main reasons behind the growing use of this technique are the recent development of low-cost recorders [9,10] and technical innovations in acoustic data processing, such as BirdNET [11,12]. 

Passive acoustic surveys generate a significant amount of data that present challenges for audio interpretation, and most projects require the use of automated signal recognition software (e.g., [11,13,14]. Birds constitute the group most commonly studied via PAM [7]; consequently, numerous recent studies have improved techniques and analyses on the basis of the automatic acoustic recognition of birds [2]. Among these tools, the recently introduced free machine learning tool BirdNET is worthy of attention. BirdNET employs deep neural network algorithms for the automated detection and classification of 6500 wildlife species [11,15]. BirdNET algorithms are trained via vocalizations from various species, including mainly birds but also some amphibians and primates (see [16,17,18]). One of BirdNET’s key advantages over other automated detection software is that the recognizers are readily available, eliminating the need for advanced computer programming skills, and can be easily operated via graphical interfaces on Windows platforms, thus avoiding the complexity of programming languages such as R or Python. However, our current knowledge about the ability of BirdNET to monitor tropical birds is very limited, without any case studies published yet (reviewed by [15]). 

Therefore, in this study, we aim to assess, for the first time, the effectiveness of BirdNET in identifying three Neotropical passerine birds and to utilize this tool to gain new insights into their ecological behavior. More specifically, we aimed to (1) evaluate the precision of BirdNET in correctly identifying the vocalizations of three closely related Neotropical passerines; (2) determine the optimal confidence threshold for each species, ensuring that BirdNET predictions can be filtered to remove predictions with low confidence; and (3) use BirdNET over a dataset of acoustic recordings collected over an entire year across five acoustic monitoring locations in the Brazilian Pantanal to characterize the diel and annual patterns of vocal activity of the studied species. This study aims to enhance the quality of passive acoustic research via acoustic sensors and the BirdNET algorithm. Additionally, our findings will improve the understanding of the ecology of tropical birds and the seasonal dynamics within the Brazilian Pantanal, the largest wetland in the world.

## 2. Materials and Methods

### 2.1. Study Species

In this study, we used Buff-breasted Wren (*Cantorchilus leucotis*), Moustached Wren (*Pheugopedius genibarbis*), and Thrush-like Wren (*Campylorhynchus turdinus*) as target species. The three species are cataloged as “Least Concern” by the Red List of the International Union for Conservation of Nature (IUCN). We selected these three species of the Troglodytidae family because they are common birds in the Neotropics, are well distributed in the Brazilian Pantanal (see next section), and are included in the latest version of BirdNET (v 2.4., [11]). Our current knowledge regarding the vocal behavior of these three species is very limited, especially with respect to Thrush-like Wren and Moustached Wren. Indeed, members of the same family will allow for comparisons of whether the vocal activity patterns of closely related species are similar. Furthermore, as wetland species, they serve as prime examples of organisms inhabiting ecosystems that are logistically challenging to monitor because of their often damp and marshy ground and typically dense but delicate vegetation [10,19].

The Buff-breasted Wren is an insectivorous and resident species typically observed in pairs in dense tangles, inhabiting gallery and riverside forests and preferring humid areas close to bodies of water [20,21]. Both sexes vocalize, and their main song exhibits frequency modulation, which is described as a sequence from one to four syllables of “wop” or “weeoh”. These songs are emitted as a duet [22].

Moustached Wrens are also insectivorous and resident, living in the dense understory of humid forests and forest edges [23]. Its song is also normally emitted in duets [24], which are characterized by a series of quick, happy phrases that are frequently repeated and are sometimes followed by a quick “cho cho cho” [23].

The resident and insectivorous Thrush-like Wren has a cooperative character and is usually found in groups [25]. They primarily inhabit the canopies of humid forests, including disturbed areas [26]. Its song is described as a variable number of “chuk, chuk, chu-rú” [25].

### 2.2. Study Area

The study was conducted in the northeastern part of the Brazilian Pantanal (Pantanal matogrossense) and included five acoustic monitoring stations located near the SESC Pantanal complex (Mato Grosso, Brazil; 16°30′ S, 56°25′ W), separated by distances ranging from 430 to 1914 m. The Brazilian Pantanal is the largest wetland in the world, with a flooded area of 140,000 km^2^. The acoustic monitoring stations were within a mosaic of forested and savanna areas, which represent the dominant vegetation in the Brazilian Pantanal and potential habitat of the three target species [27]. It is a flat area with altitudes ranging from 80 to 100 m, an average annual temperature of 24 ℃, and an average annual rainfall ranging between 1000 and 1400 mm and is distributed seasonally [28]. The climate is tropical and humid [13,27].

The study area was located within the alluvial plain of the Cuiabá River, one of the main tributaries of the Paraguay River within the Pantanal [27], which in turn is one of the main tributaries of the Paraná River [29], with a drainage area of 280,000 km^2^ [30]. This plain is characterized by seasonal floods, which cause transitions from terrestrial to aquatic habitats and vice versa [31]. These floods are due to seasonal rainfall occurring between October and April [27], during which 80% of the Pantanal is flooded [32] because of the reduced runoff capacity of the drainage basin. The dry season occurs from May to September [27], when water is lost through evaporation and infiltration [33]. Because of these seasonal changes, the use of noninvasive techniques, such as PAM, can better contribute to wildlife monitoring.

### 2.3. Recording Protocol

The acoustic monitoring stations operated daily from 8 June 2015 to 31 May 2016, covering an annual cycle at each site. The locations in which they were placed were selected to encompass the most representative plant formations of the Brazilian Pantanal (forests and savannahs). A Song Meter SM2 recorder (Wildlife Acoustics, Maynard MA, USA) was placed at each station. The recorders were programmed to record (.wav format) the first 15 min of each hour 24 h a day with a sampling frequency of 48 kHz and a resolution of 16 bits per sample [13]. Recorders were checked approximately every two weeks to download data and change batteries.

### 2.4. Acoustic Data Analysis

BirdNET segments recordings into 3-second intervals, extracting signal characteristics and detecting matches with its model of singing patterns; it reports detections accordingly [11]. Moreover, BirdNET can identify multiple species within the same segment and provides a quantitative confidence score for each detection, ranging from 0 to 1. This score reflects the probability of accurately identifying the species, with a score of 1 indicating a near-perfect match to BirdNET’s understanding of the species [12,34]. The users of BirdNET can adjust a threshold value to filter application results on the basis of their desired confidence level. Optimizing for higher confidence values increases the accuracy percentage of correct detections relative to all detections considered but might also reduce the total number of detections. This can significantly reduce the number of false positives but can increase the number of false negatives. However, there is currently a limited understanding of how confidence values affect the accuracy of BirdNET species detection (reviewed by [15]).

Once the recordings were completed, they were analyzed via the “Multiple Files” tab in the GUI interface of BirdNET-Analyzer (version 2.4, https://github.com/kahst/BirdNET-Analyzer, accessed on 12 August 2024) [11] We used the default values, which were as follows: confidence threshold of 0.1, sensitivity parameter of 1.0, and no overlap (0 s). We applied a “Custom Species List” filter to configure BirdNET to detect only the three target species, thus avoiding the detection of nontarget species [10,12]. BirdNET was programmed to process one recording at a time via four computer threads. The total scanning time was approximately 142 h (2.3% of the total recording time).

### 2.5. BirdNET Performance Evaluation

To evaluate the effectiveness of BirdNET in detecting the three study species, the detection accuracy was estimated for each species separately. Accuracy was assessed without applying any confidence threshold filtering and was defined as the percentage of correctly identified predictions out of the total predictions reviewed [35]. A sample of 450 predictions was randomly selected from the BirdNET output for each species by considering 50 predictions for each 0.1 confidence score class (i.e., 50 predictions with confidence scores ranging between 0.1 and 0.2, 50 between 0.2 and 0.3, etc.). For each prediction, an experienced observer listened and visually inspected the audio spectrogram at the timestamp of the 3-second segment reported by BirdNET in the free software Audacity (v 2.3., [36]) and verified whether the target species was present or absent. The BirdNET precision was estimated (in %) by dividing the number of BirdNET predictions correctly classified by the total number of BirdNET predictions verified.

### 2.6. Statistical Analyses

The 450 predictions verified for each species were also used to estimate the confidence score threshold with a 90% probability of correct identification for each species. This estimate allowed us to filter the BirdNET output by removing predictions with low confidence scores and deriving ecological results (see first application in birds’ vocal activity in [12]). We opted for 90% confidence to keep a high number of predictions and because prior research has found no notable differences when describing singing patterns using high, although variable, confidence scores [17]. We followed the approach outlined in [12] (see also [17] for first application in anurans), so we back-transformed BirdNET’s confidence scores into its original logit scale. Then, for each of the three species, we fitted a logistic regression using the correct or incorrect classification of the verified predictions as a response variable and the BirdNET logit-scale prediction score as the independent variable. The logistic regressions provided an equation that enabled us to convert BirdNET scores into the probability of a given prediction being correct. For each species, the equations considering a probability of correct identification of 90% were as follows:Threshold = (ln (p/(1 − p)) − α)/β,(1)
where p is the threshold selected (0.90 in our case), α is the intercept of the logistic regression, and β is the slope of the regression.

The identified optimal score was used as a confidence score threshold to finally consider only BirdNET predictions with a high probability of correct identification when describing the diel and seasonal patterns of vocal activity of the three monitored species. The patterns of vocal activity were described by pooling the data from the five acoustic monitoring stations.

## 3. Results

### 3.1. BirdNET Performance

The BirdNET precision slightly varied among the three species (Table 1). The lowest precision was reached for the Moustached Wren and the Buff-breasted Wren, for which 326 and 344 of the 450 BirdNET predictions verified for each species were correctly classified (precision of 72.4% and 76.4% for the Moustached Wren and the Buff-breasted Wren, respectively, Table 1). The highest precision was reached for the Thrush-like Wren (84% precision, Table 1), for which 378 of the 450 BirdNET predictions verified were correct. The probability of BirdNET correctly classifying a bird vocalization varied depending on the confidence value of the predictions, with greater precision at higher confidence values. For example, the average BirdNET precision for the three target species at confidence score values between 0.1 and 0.5 was 61% (366 bird vocalizations correctly detected among 600 BirdNET predictions verified), whereas at confidence scores above 0.5, the average BirdNET precision was 90.9% (682 of the 750 BirdNET predictions verified correctly classified, Table 1).

After the logistic regressions were fit, the minimum confidence score to consider only detections with a 90% probability of correct identification was 0.603 for the Buff-breasted Wren (Figure 1A) and 0.428 for the Thrush-like Wren (Figure 1B). Owing to the lower precision of BirdNET for correctly detecting the Moustached Wren, especially at high confidence score intervals (see Table 1), it was impossible to identify an optimal confidence score that was able to correctly predict the vocalization of the species; therefore, its vocal behavior was not described.

### 3.2. Vocal Activity Patterns

After applying the confidence thresholds, the sample size used for the description of the song patterns was 13,612 vocalizations for Thrush-like Wren and 491 for Buff-breasted Wren. To facilitate reading, hereinafter, we use the term vocalization when referring to the BirdNET predictions filtered. The daily vocal activity patterns of both species were very similar (Figure 2). Both the Thrush-like Wren and the Buff-breasted Wren exhibited a bimodal vocal activity pattern, with peaks around sunrise and sunset and low vocal activity during the central hours of the day and almost none during the night (Figure 2). The largest peak vocal activity of both species occurred during the three hours after sunrise, with over 50% of the total vocal activity recorded between 6 a.m. and 8 a.m. (see detailed tables of the hourly vocal activity of each species per station in Appendix A (Table A1 and Table A2).

The Buff-breasted Wren and the Thrush-like Wren exhibited similar vocal behavior patterns, with peaks in vocal activity occurring between March and June during the onset of the dry season (Figure 3). The percentage of vocalizations detected between March and June, relative to the total, was 43.6% for the Thrush-like Wren and 46.0% for the Buff-breasted Wren. However, both species presented secondary vocal activity peaks during the remainder of the year, especially in December (12.0% of the total for the Buff-breasted Wren and 8.9% for the total for the Thrush-like Wren). Overall, both species displayed similar singing activity patterns throughout the year. Furthermore, both the Buff-breasted Wren and the Thrush-like Wren were detected throughout the entire annual cycle (Figure 3). Detailed tables of the monthly vocal activity of each species per station can be found in Appendix A (Table A3 and Table A4).

## 4. Discussion

In this study, we validated, for the first time, the use of acoustic sensors coupled with BirdNET, a free-to-use and user-friendly machine learning tool, for detecting and studying the ecology of tropical birds. The mean precision of BirdNET for correctly identifying the three target species was similar and high (range 72–84%). However, owing to variations among species in the ability of BirdNET to correctly classify their vocalizations at high confidence scores (Table 1), it was impossible to estimate an optimal confidence score, which aimed to filter BirdNET output and retain only predictions with a high probability of being correct (>90%), for the Moustached Wren. We were able to estimate such an optimal confidence score threshold for the other two species, which allowed us to describe their vocal activity patterns using only BirdNET predictions with a high probability of being correctly identified. We are aware that the definition of what is an optimal confidence score threshold may vary among studies; therefore, users may select one or another threshold according to their research goal. For example, a low confidence score threshold may be selected and followed by output verification, if the aim is to detect the presence of threatened or invasive species, to facilitate effective management (e.g., [18,37]), whereas the aim to describe vocal activity patterns may be enough to select a high confidence score threshold without further output verification (e.g., [12,17]). When the average precision obtained in this study for the three target species (77.6%) was compared with the average precision for 984 species of European and North American birds (79.0%) [11], we observed that the precisions for the three tropical birds were very similar. This result is even more surprising considering that the precision estimated for European and North American birds was calculated via focal recordings and therefore collected with high-quality directional microphones, whereas our recordings were collected with omnidirectional microphones.

Various authors have suggested that to work with most species through BirdNET, it is appropriate to use confidence score threshold values greater than 0.5 [15,38] and even values that range between 0.7 and 0.8, as these values yield the greatest number of correct identifications (95% probability of correct identification) [39]. However, prior research, using the same approach as in our study, identified optimal confidence scores lower than 0.5 to retain only predictions with high probabilities of being correctly identified (see [12]), which suggests the need for species-specific research. Our findings also suggest that estimating an optimal confidence score might not be possible for certain species but might be highly variable even among closely related species (0.603 for the Buff-breasted Wren and 0.428 for the Thrush-like Wren). These findings are in agreement with prior research proving the existence of large variations in optimal confidence scores among studies for the same species (see, e.g., [39,40,41]). Indeed, the effectiveness of BirdNET correctly identifying bird vocalizations may vary among areas and periods of the day due to variations in ambient noise (e.g., [42]). Further research should explore the reasons behind these variations in BirdNET precision among species and among studies. In the meantime, we must be cautious when extrapolating the precision and optimal confidence scores between species and/or studies [15]. Likewise, further research should assess the ability of BirdNET to detect bird vocalizations with an acoustic metric known as the recall rate, which is not frequently evaluated in BirdNET surveys [15].

Both the Buff-breasted and the Thrush-like Wren showed a bimodal vocal activity pattern, in line with the daily patterns described for most bird species (reviewed by [43]), including other Neotropical wrens (e.g., *Campylorhynchus rufinucha*, [44]) and other songbirds in the study area (e.g., [42]). Both species presented the largest peak of vocal activity around dawn and a second, lower peak around dusk. Our results agree with prior descriptions of the Buff-breasted Wren’s daily singing pattern, with pairs singing together near sunrise, a decrease in activity during the late morning and afternoon, and a second peak of singing activity in the evening [20]. This second peak of activity has been proposed to improve communication between members of the same pair, with both mates vocalizing in the evening before entering their nests or roosts [22]. Our findings constitute the first description of the daily singing pattern of the Thrush-like wren, which overall agrees with the results reported for other Neotropical wrens ([44], Figure 2). However, our current knowledge regarding the factors influencing diel vocal activity patterns of tropical birds is very limited (but see, e.g., [45,46,47,48,49]).

Both species also presented very similar seasonal patterns of singing activity, with relatively constant vocal activity throughout the year but with a relatively large peak of singing during the period of March–June. The detection of both species over a complete year confirms their status as resident species in the Brazilian Pantanal [50,51]. Both species are territorial, and their songs have been proposed to serve multiple functions, including as a territorial defense [20,44], which may contribute to explaining the relatively consistent, although a low pattern of singing observed throughout the year. Research has shown that seasonal changes in water levels affect bird ecology, including nesting, feeding, and vocal patterns [31,52,53,54,55]. The Brazilian Pantanal experiences annual floods, resulting in seasonal variations in terms of insect abundance and diversity [56]. Therefore, the peak of singing activity between March and June is likely related to changes in insect abundance in response to the pronounced seasonality of the Brazilian Pantanal region. For example, studies indicate peaks in the abundance of several dipteran species and ants at the end of the wet season and start of the dry season, typically between May and July, driven by climatic conditions [57,58], with ants being a crucial food resource for many Neotropical insectivorous passerines [59]. Overall, the increased food availability following water recession may stimulate the reproduction of both wrens during this period and therefore explain the peak of singing activity since songs of Neotropical wrens are also used for mate attraction and pair bonding [20,44].

The proposed periods for breeding, which are based on seasonal changes in singing activity, require further research, together with field observations, to characterize the breeding ecology of the target species better and to assess whether their nesting behavior is influenced by the flood dynamics of the Pantanal. Indeed, we cannot rule out that the observed decrease in vocal activity during the wet season might be associated with a lower amount of time spent near the recorders as flooded areas shrink, making food searches more challenging [60,61]. Moreover, diminished vocal activity during the rainy season might result from inefficiencies caused by rain masking vocalizations [62,63] or individuals seeking shelter [64].

## 5. Conclusions

We provide the first assessment of the effectiveness of acoustic sensors coupled with BirdNET, a new tool for processing acoustic data, for correctly identifying and monitoring the vocal activity of three Neotropical passerine birds. The precision of BirdNET for correctly identifying the three species was high, although it was variable. Indeed, it was not possible to identify an optimal confidence score threshold for one of the species. The optimal confidence scores identified in that study, although valuable as starting points for further research with the target species, should be assessed if they are to be applied in different regions. The other two wrens considered showed similar diel and seasonal patterns of singing activity, with increased vocal output around the crepuscular periods and during the dry season, when insect abundance and availability might be high. Further research should evaluate the performance of BirdNETs for monitoring a broader range of tropical species, including assessments of the ability of BirdNET to detect bird vocalizations (recall rate). We hope that this study will encourage researchers and managers to utilize this readily available tool to generate valuable scientific data. The use of acoustic sensors, coupled with BirdNET, might be especially useful for improving our current knowledge regarding the ecology of tropical and wetland birds, which are species for which there is limited knowledge and challenging monitoring.

## Figures and Tables

**Figure 1 sensors-24-05780-f001:**
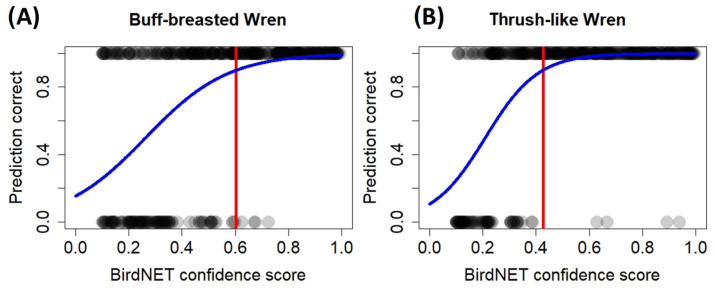
The results of the logistic regression (blue line) showing the relationship between the probability of a correct BirdNET prediction and the confidence score of a given prediction for the (**A**) Buff-breasted Wren and the (**B**) Thrush-like Wren. Statistical analyses were performed using the BirdNET logit-scale of the prediction score as an independent variable, but we represent the original confidence score of BirdNET for graphical purposes. The red solid lines show the optimal confidence score threshold identified for each species.

**Figure 2 sensors-24-05780-f002:**
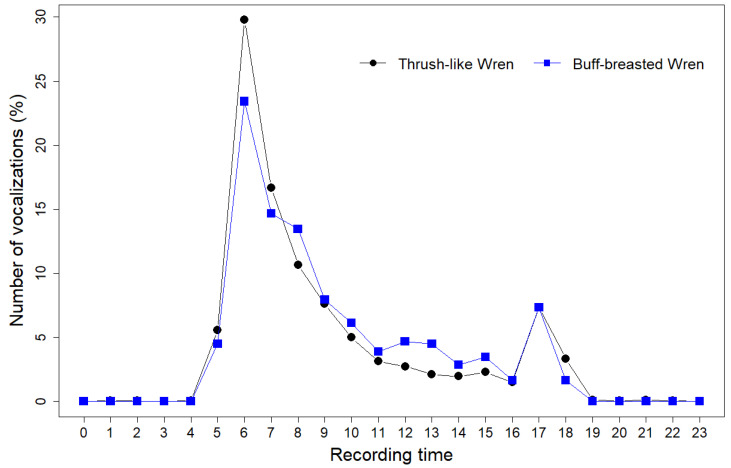
The daily pattern of vocal activity of the Thrush-like Wren (black) and the Buff-breasted Wren (blue) in the Brazilian Pantanal. The daily pattern of vocal activity refers to the percentage of vocalizations above the optimal confidence score detected per hour for each species. Times are expressed in terms of local winter time (UTC-4) and number.

**Figure 3 sensors-24-05780-f003:**
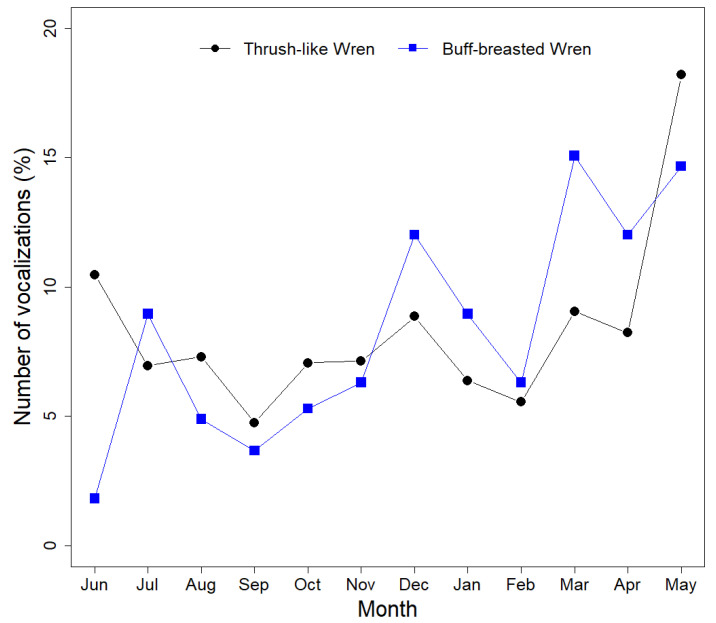
The annual pattern of vocal activity of the Buff-breasted Wren (blue) and Thrush-like Wren (black) in the Brazilian Pantanal. The annual pattern of vocal activity refers to the percentage of vocalizations above the optimal confidence score detected per month for each species. Monitoring was performed via passive acoustic monitoring from 8 June 2015 to 31 May 2016.

**Table 1 sensors-24-05780-t001:** The number of BirdNET predictions correctly classified and the BirdNET precision (in %) for detecting three Neotropical passerines. The values are shown separately for each species and for the following confidence score interval classes: 0.1–0.3; 0.3–0.5; 0.5–0.7; >0.7; and for the whole range, 0.1–1. For each confidence interval class, a total of 100 predictions were verified for each species, except for the class > 0.7, for which 150 predictions were verified.

Species	Predictions	0.1–0.3	0.3–0.5	0.5–0.7	>0.7	0.1–1
Moustached Wren	Correct	60	66	87	113	326
	Precision	60%	66%	87%	75.3%	72.4%
Buff-breasted Wren	Correct	42	66	87	149	344
	Precision	42%	66%	87%	99.3%	76.4%
Thrush-like Wren	Correct	47	85	98	148	378
	Precision	47%	85%	98%	98.7%	84%

## Data Availability

Raw databases employed for describing birds’ vocal behavior are published as the Appendix A. The number of vocalizations detected per hour and month at each acoustic monitoring station for each of the three species can be found in the Table A1, Table A2, Table A3 and Table A4.

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
