# Peer review of "Living Together, Singing Together: Revealing Similar Patterns of Vocal Activity in Two Tropical Songbirds Applying BirdNET"

_sensors, 2024, doi:10.3390/s24175780_

Round 1

Reviewer 1 Report

Comments and Suggestions for Authors

In this paper, PAM was used for sound acquisition, and BirdNET was used for recognition and classification. The research was interesting, but the reviewers had the following questions.

How do we evaluate the accuracy of BirdNET's algorithm under different ambient noise conditions? Is there data to support its performance in a specific ecological environment like the Brazilian Pantanal?

Although the BirdNET algorithm showed acceptable accuracy in this study, did the study evaluate the algorithm's ability to generalize to untrained sounds?

Why were these three specific tropical songbirds chosen for study? Is there any special ecological significance? What is their protection status?

What criteria are used for the placement of recording equipment and the selection of monitoring sites? Is it possible that there is selection bias that affects the representation of the sound data?

Does the study fully consider the impact of sample size on algorithm accuracy? Although the study collected a large amount of data in one year, were all time periods and environmental conditions adequately represented? For example, is there a balanced amount of data collected during the rainy and dry seasons, or during different time periods (such as early morning and dusk)? The unbalance of samples may affect the algorithm's accurate recognition of bird activity patterns in different time periods.

How to explain the peak singing activity of the studied species between March and June? Are other possible environmental factors, such as rainfall or changes in vegetation, taken into account?

What method did the authors use to determine the confidence threshold for a 90% correct identification probability? Is it possible that this threshold needs to be adjusted for different species?

Can the findings be generalized to other tropical birds or other regions? Are there any plans for cross-regional or cross-species comparative studies?

The limitations of this paper should be highlighted in the conclusion. How should future research overcome these limitations and further leverage BirdNET's algorithm to improve the efficiency and accuracy of ecological monitoring?

It is recommended that academic papers should be written and described in a third-person objective tone rather than a first-person subjective attitude.

Comments on the Quality of English Language

In this paper, PAM was used for sound acquisition, and BirdNET was used for recognition and classification. The research was interesting, but the reviewers had the following questions.

How do we evaluate the accuracy of BirdNET's algorithm under different ambient noise conditions? Is there data to support its performance in a specific ecological environment like the Brazilian Pantanal?

Although the BirdNET algorithm showed acceptable accuracy in this study, did the study evaluate the algorithm's ability to generalize to untrained sounds?

Why were these three specific tropical songbirds chosen for study? Is there any special ecological significance? What is their protection status?

What criteria are used for the placement of recording equipment and the selection of monitoring sites? Is it possible that there is selection bias that affects the representation of the sound data?

Does the study fully consider the impact of sample size on algorithm accuracy? Although the study collected a large amount of data in one year, were all time periods and environmental conditions adequately represented? For example, is there a balanced amount of data collected during the rainy and dry seasons, or during different time periods (such as early morning and dusk)? The unbalance of samples may affect the algorithm's accurate recognition of bird activity patterns in different time periods.

How to explain the peak singing activity of the studied species between March and June? Are other possible environmental factors, such as rainfall or changes in vegetation, taken into account?

What method did the authors use to determine the confidence threshold for a 90% correct identification probability? Is it possible that this threshold needs to be adjusted for different species?

Can the findings be generalized to other tropical birds or other regions? Are there any plans for cross-regional or cross-species comparative studies?

The limitations of this paper should be highlighted in the conclusion. How should future research overcome these limitations and further leverage BirdNET's algorithm to improve the efficiency and accuracy of ecological monitoring?

It is recommended that academic papers should be written and described in a third-person objective tone rather than a first-person subjective attitude.

Reviewer 2 Report

Comments and Suggestions for Authors

General comments

I have no major concerns with this paper, however I do feel that it is somewhat lacking because it fails to include measures of recall. Understanding false negatives is often as important (sometimes more important) as understanding precision when considering tools like BirdNET. For example, if a recognizer misses most of a species’ calls at a high score threshold, that would reduce the reliability of the data for any subsequent analyses like vocal activity over a day. Ideally, the authors would measure recall for a subset of data. If not, as a minimum, the introduction and discussion should address the issue. What is precision and recall? When might one be more important than the other? How are they measured? What does the literature say about their trade-off? Etc. Knight et al (2017), which is cited in this manuscript, is good place to start.

Abstract

L23 and elsewhere: Is this really the first time BirdNET has been assessed for tropical songbirds? I assume, as a minimum, the developers of BirdNET assessed performance in some capacity when training the model/s?

L24 – 27: Confusing that three species are mentioned, but only two are named. I suggest clarifying this. On first reading, I thought ‘three’ was an error.

L28: Maximum of what?

Introduction

L37: Remove ‘that are able’

L39 – 40: Monitoring doesn’t necessarily ‘ensure’ proper management. I suggest re-wording this.

L41: Methods for birds?

L47: In this study, effectiveness is only measured as precision, correct? It warrants some discussion and justification, given the various performance metrics for call recognizers.

L56: “…, passive acoustic monitoring (PAM), has emerged (Sugai…”

L58: Time periods?

L60: It might be automated, but it might not be. Often we’d consider it semi-automated – using a call recognizer and then verifying some or all of the outputs.

L67: These days I’d say that most projects need some form of signal processing. Manual methods are rarely used now.

L73: 6,500 individual vocalisations? Or 6,500 species? What specifically is this value referring to? Unclear.

L87: Does the ‘optimal’ confidence threshold depend on the objective of the project? Given that you have defined a precision target of 90%, I suggest being more specific here.  Optimal can mean different things to different people/projects.

Methods

L162: Please refer to Wood and Kahl (2024), Journal of Ornithology, for a discussion on what these confidence scores are and are not. This paper explains that raw scores are not probabilities. Given your manuscript is focussed largely on confidence scores, Wood and Kahl (2024) should definitely be read and cited.  

L165 – 167: Wording is unclear. Maybe split into two sentences (e.g. “… total number of detections. This can significantly reduce the number of false positives, but can increase the number of false negatives”).

L183 – 184: Knight et al (2017) uses three metrics for evaluating recognizer performance: precision, recall and F-score. It is unclear why this paper has chosen to calculate only precision. This requires justification, as a minimum. Better still, I suggest directly calculating recall for a subset of your data.

L192: Do you mean 450 vocalizations?

Results

L226 – 227: This makes sense but, again, what does this mean for recall?

L253: Wording.

Fig 2: Is the y-axis the % of total vocalizations detected/verified for that species?

Discussion

L289: As elsewhere, ‘optimal’ is subjective and depends on the aim of the work, correct?

L299: Focal what? Recordings?

Round 2

Reviewer 1 Report

Comments and Suggestions for Authors

I have no other comments.

Comments on the Quality of English Language

I have no other comments.